# Underweight, Overweight and Obesity among Reproductive Bangladeshi Women: A Nationwide Survey

**DOI:** 10.3390/nu13124408

**Published:** 2021-12-09

**Authors:** Mansura Khanam, Uchechukwu Levi Osuagwu, Kazi Istiaque Sanin, Md. Ahshanul Haque, Razia Sultana Rita, Kingsley Emwinyore Agho, Tahmeed Ahmed

**Affiliations:** 1International Centre for Diarrhoeal Disease Research, GPO BOX 128, 68, Shaheed Tajuddin Ahmed Sarani, Dhaka 1212, Bangladesh; sanin@icddrb.org (K.I.S.); ahshanul.haque@icddrb.org (M.A.H.); rita@icddrb.org (R.S.R.); tahmeed@icddrb.org (T.A.); 2Translational Health Research Institute (THRI), School of Medicine, Western Sydney University, Campbelltown, NSW 2560, Australia; L.Osuagwu@westernsydney.edu.au (U.L.O.); K.Agho@westernsydney.edu.au (K.E.A.); 3School of Health Sciences, African Vision Research Institute (AVRI), University of KwaZulu-Natal, Westville Campus, Durban 3629, South Africa; 4School of Health Sciences, Western Sydney University, Locked Bag 1797, Penrith, NSW 2751, Australia

**Keywords:** obesity, nutrition, women, body mass index (BMI), Demographic Health Survey (DHS)

## Abstract

The double burden of malnutrition is becoming more prevalent among Bangladeshi women. Underweight, overweight, and obesity were examined among women aged 15–49 years using the 2017–2018 Bangladesh Demographic and Health Survey (BDHS). A dataset of 20,127 women aged 15–49 years with complete Body Mass Index (BMI) measurements were extracted and categorized as underweight, normal weight, overweight, and obesity. A multiple logistic regression that adjusts for clustering and sampling weights was used to examine underweight, overweight, and obesity among reproductive age Bangladeshi women. Our analyses revealed that the odds of being overweight and obese were higher among women who completed primary and secondary or more levels of education, rich households, breastfeeding women, and women exposed to media (newspapers and television (TV). Women from the poorest households were significantly more likely to be underweight (AOR = 3.86, 95%CI: 2.94–5.07) than women from richer households. The likelihood of being underweight was higher among women with no schooling, adolescent women, and women not using contraceptives. Conclusions: Overweight and obesity was higher among educated and affluent women while underweight was higher among women from low socioeconomic status, indicating that tailored messages to combat overweight and obesity should target educated and affluent Bangladeshi women while improving nutrition among women from low socioeconomic status.

## 1. Introduction

In the last three decades, the average body mass index (BMI) has risen by about 0.4 kg/m^2^ annually [1], with greater increases in overweight and obesity rates in developing countries [2,3,4]. The double burden of nutritional status has been used to refer to underweight and overweight and is now recognized as a global epidemic [5,6,7]. Like many Southeast Asian developing economies, Bangladesh is confronted with this double burden of continually high undernourishment, and an increase in overweight and obesity [8,9]. Although the risk of infectious diseases is higher among underweight individuals, public health concerns due to diseases caused by overweight and obesity are growing faster because they have significant and adverse health and socioeconomic consequences, both immediate and long-term [2].

In Bangladesh, there is a high prevalence of maternal malnutrition, especially among adolescent girls [4], and this contributes to an intergenerational cycle of malnutrition and poverty. Layered on top of this is the unacceptably high prevalence of anaemia among adolescent girls and pregnant women, affecting up to 50% of pregnant women and 40% of non-pregnant women in Bangladesh [4]. It has been reported that 55% of non-pregnant women are zinc deficient, with another 22% of vitamin B12 deficient women [10]. Although between 1997 and 2007, there has been some improvement in the nutritional status of Bangladeshi women (indicated by reduction in the proportion with BMI < 18.5 kg/m^2^) from 52%–30% [8], at the same time, there has been an upward trend in the prevalence of overweight and obesity and a gradual decrease in the proportion of underweight women [1,6,7,9,11]. Similarly, the obesity and overweight trends were reported in India [12], were higher in adult females than males in Bangladesh [5,6], and were responsible worldwide for approximately 3.8 percent of the life of disability-adjusted [13,14,15]. This increased prevalence has been linked to rapid urbanization, reduced physical activity, and changes in lifestyle and dietary consumption [16,17,18,19]. 

Underweight and overweight could be linked to premature birth and other adverse results [20]. In developing countries, underweight and undernourishment are predominant among women due to poverty, food scarcity, and illiteracy [21,22,23]. Women of reproductive age (WRA) who are underweight (BMI < 18.50 kg/m^2^) have an increased risk of low birth weight, intrauterine growth constraint, neonatal morbidity, mortality, and a lack of growth [22,24,25]. The positive nutrition shift in Bangladesh has contributed to a reduction in underweight status through the microfinance schemes, the growing gross domestic product and healthy diet, access to medical care, and more women’s education [3,22,24,25,26,27].

A secondary analysis of BDHS 2014 showed that around 23% of WRA in Bangladesh were overweight and obese [2], with higher rates among urban women [11]. This could lead to an increase in the burden of non-communicable diseases, including diabetes and cardiovascular diseases [28]. Current research on malnutrition in Bangladesh did not consider overweight and obesity, which are expected to be more prevalent in affluent urban areas [7]. In addition, there are reports that overweight and obesity rates continue to increase among WRA in Bangladesh and rise from 11.8% in 2007 to 23.8% in 2014 [2]. In a recent study, underweight and overweight factors were examined for women of reproductive age in Bangladesh using BDHS 2014. The authors utilized a multinomial logistic regression model and showed that relative to normal weight, higher odds for overweight and obesity in Bangladeshi were significantly associated with older women, higher parity, higher education, frequently watching TV, urban residence, and affluent communities [2].

Additionally, the prevalence of overweight and obesity rose by 8.8% and 29.9% in 2017–2018 BDHS [29] and was 24% in 2014 BDHS [5]. However, it is crucial to understand whether the same factors continue to drive the overweight and obesity rates in women of reproductive age in Bangladesh. This will enable targeted nutritional policies that could improve the outcome of the women in Bangladesh because, overweight and obesity play an increasingly important role in reducing maternal mortality because of increased risk of hypertension, pre-diabetes, type 2 diabetes, dyslipidemia, cardiovascular diseases [30], which have severe consequences to mothers and their offspring’s.

Therefore, this study will explore the determinants related to underweight, overweight, and obesity among reproductive age (15–49 years) women using the recent 2017–2018 BDHS. These study results will allow researchers and policymakers to review and redesign new community-based public health intervention strategies to reduce excess weight and obesity among females and children of reproductive age. The paper further examines the nutritional gap by identifying the target group of women in Bangladesh with both underweight and overweight status.

## 2. Materials and Methods

### 2.1. Data Sources

The 2017–2018 BDHS represents a national survey because it encompasses the whole population living in Bangladeshi non-institutional housing units. The BBS 2011 sampling framework used in the survey included enumeration areas (EAs) of the 2011 Population and Housing Census, which are provided by the Bangladesh Bureau of Statistics (BBS). The Primary Sampling Unit (PSU) (i.e., clusters) for the survey includes an EA of about 120 households on average. Each cluster was considered as a community based on previous studies [2,31].

Bangladesh is made up of eight divisions, including Barishal, Chattogram, Dhaka, Mymensingh, Khulna, Rajshahi, Rangpur, and Sylhet. There are zilas (districts) for each division, and each zila is subdivided further in Upazilas (sub-district). Each urban area of Upazila is split into wards, further divided into Mohallas, whereas each rural area in Upazila is separated into parishes of union (UPs). There are Mouzas in UPs, and all these divisions enable the division into rural and urban areas. Figure 1 presents the sampling procedure. The survey included a stratified two-stage sample of households: 675 EAs with probability proportions to the size of EA were selected in the first phase. In urban areas, there were 250 EAs, and in rural areas, 425 EAs. The sample was drawn by BBS in the first stage, in accordance with the specifications of the DHS team. The selection resulted in 20,250 residential households in accordance with this design. Approximately 20,100 married women aged 15–49 years are expected to complete interviews. The survey report contains details of the sample structure, including the sample framework and the sample implementations [5]. 

Weight measurement using the lightweight scale SECA787 (with digital screens). Heights were measured using an adjustable wood measuring board, designed specifically to provide an accurate reading of 0.1 cm to take the developed countries into account. Through weight and height measurements, their BMI was calculated. The survey also included information on fertility, contraceptive use, maternal and child health, mother’s nutritional status, women’s empowerment, and sociodemographic characteristics.

### 2.2. Exclusion Criteria

Data for 20,127 women 15 to 49 years of age who were not pregnant have been used following exclusion. 

### 2.3. Dependent Variables

The dependent variables were ordered as normal, underweight, overweight, and obesity which was based on the WHO classes for BMI: underweight (<18.50 kg/m^2^), normal (18.5–24.9 kg/m^2^), overweight (25.0–29.9 kg/m^2^), and obesity (≥30.0 kg/m^2^) (48). In order to ensure the quality, all weight/height were recorded, BMI has been continuously extended in conservative form.

### 2.4. Independent Variables

The independent variables were the individual-, household- and community-level factors identified in the conceptual framework. The individual-level variables included all relevant attributes of the respondents, including maternal work status, parents’ level of education, mother’s marital status, mother’s age, mother’s literacy status, access to health care services (autonomy to health care), access to the media (newspaper, radio and Television (TV)), and power over family income. Household-level variables consisted of the source of drinking water (improved or unimproved) and household wealth index into five categories [poorest, poorer, middle, richer, and richest]. In creating a wealth index [32], principal component analysis was used to estimate the index weights based on acquired information on various household assets, including ownership of different means of transport and other sustainable domestic goods. This index was divided into five categories, and one of five categories was allocated to each household. Variables at the community level included residence (urban/rural) and geographical area (Barishal, Chattogram, Dhaka, Khulna, Mymensingh, Rajshahi, Rangpur, and Sylhet).

### 2.5. Statistical Analyses

All analyses were conducted using Stata version 14.1 (Stata Corp 2015, College Station, TX, USA). The command ‘Svy’ has been used for the adaptation of the cluster sampling design, weights, and the Taylor series linearized procedure was used to calculate the standard errors. The dependent variable was always expressed as binary, with number ‘1’ assigned as underweight, overweight, and obesity, while 0 was normal weight. Frequencies or proportions were used to show the prevalence of overweight and obesity and their 95 percent confidence intervals, using descriptive statistics and surveying tabulation. Logistics regression was adjusted using the cluster and survey weights.

Multivariable logistic regression analysis was performed to obtain the association of each independent variable with the dependent variable (i.e., underweight, overweight, and obesity), utilizing a normal BMI range as the reference value. Crude and adjusted regression models were built and variables with a pre-specified significance value of <0.2 in the unadjusted model were eligible for inclusion in the final adjusted multivariable models [27]. Association results of multivariable regression analysis were presented by odds ratio (OR) at 95% confidence intervals (CIs). Statistical significance was considered with a *p*-value < 0.05. Our final model was tested for any co-linearity. The adjusted regression models’ odds ratios and the 95% confidence intervals (CI) were determined.

## 3. Results

### 3.1. Prevalence of Underweight, Normal, Overweight and Obesity among Reproductive-Age Women in Bangladesh

The overall prevalence of the different categories of BMI among reproductive-age women in this study is shown in Figure 2. About one-third of the women who participated in the 2017 BDHS survey were either overweight or obese with few underweight women (8.7%, 95% CI: 8.2–9.3%).

For both overweight and obesity, the majority of them belonged to the age group 35–49 years (29.9%, 95% CI: 28.5–27.8% for overweight and 8.8%, 95% CI: 8.0–9.7% for obesity; while the majority of those aged 15–24 years were underweight (13.3%, 95% CI: 12.3–14.5%). There was a significant increase in the prevalence of overweight and obesity and a decrease in the prevalence of underweight with higher education. Women with secondary or higher education reported the highest prevalence of overweight (29.9%, 95% CI: 28.8–31.1%) while those with no education reported the highest underweight prevalence (11.4%, 95% CI:10.3–12.7%). The majority of the women who were either overweight or obese came from the wealthiest households while underweight women were mostly from the poorest households. There was also a higher prevalence of overweight (29.8%, 95% CI: 28.7–30.9) and obesity (9.3%, 95% CI: 8.5–10.1) and a lower prevalence of underweight 6.3%, 95%CI: (5.7–6.9%) in those women that watched television at least once a week. The prevalence of overweight and obesity in Dhaka, Chattogram, Khulna, and Barishal was higher than in Sylhet (Table 1).

### 3.2. Factors Associated with Underweight among Reproductive Age Women in Bangladesh 

The factors associated with underweight in Bangladeshi women aged 15–49 years are shown in Table 2. The odds for underweight were significantly higher among women who seldom watched TV; with no education; not currently breastfeeding, formerly married women, and women not using contraceptives at the time of this study. In addition, women who lived in other Bangladeshi divisions except for Chattogram and Dhaka were more likely to be reported as underweight. The likelihood of being underweight increased significantly among women from the poorest households (AOR = 3.86,95%CI: 2.94–5.07) and adolescent women.

### 3.3. Factors Associated with Overweightamong Reproductive Age Women in Bangladesh 

The factors associated with overweight among Bangladeshi women aged 15–49 years are shown in Table 3. After adjusting for independent variables in this study, it was found that women who completed primary or secondary or more levels of education, currently married women, women from richest households, non-adolescents (women older than 18 years), breastfeeding women, women who jointly earned with their husband were more likely to be overweight. 

### 3.4. Factors Associated with Obesity among Reproductive Age Women in Bangladesh 

Table 4 shows factors associated with obesity among women of reproductive age in Bangladesh. Multiple binary logistic regression analyses showed that women who completed primary or secondary or more levels of education, women currently breastfeeding, non-adolescent women, older women (24–49 years), and working women were significantly more likely to be obese.

The odds of being obese were also high among women from the richest households, women who read newspapers or magazines at least once a week, women who watched television at least once a week, and women who jointly make decisions with husbands on health care. Women who lived in Sylhet were 49% (AOR = 0.51, 95%CI:0.37–0.71) significantly less likely to be obese than those women who lived in the Barisal Division of Bangladesh.

## 4. Discussion

This cross-sectional study used the nationally representative 2017 BDHS dataset to examine the prevalence of underweight, overweight, and obesity among Bangladeshi women of reproductive age. Furthermore, we identified the risk factors for underweight, overweight, and obesity among these women. The findings showed that approximately two in every three women of reproductive age in Bangladesh were reported as underweight, one in every four was overweight, and about seven percent were obese at the time of data collection. That overweight and obesity rates had increased by about 8% in the last three years (from 24% in 2014 [2] to 32% in 2017) and was almost double compared with the rate in the last ten years, is an indication of a growing burden of malnutrition in form of over-nutrition among the young women in Bangladesh and these have serious consequences during pregnancy and childbirth [33,34]. The factors associated with overweight or obesity in this population included the women’s education, increasing age, and family wealth index and were identified as the drivers of overweight and obesity in this population. There were other factors such as higher education, breastfeeding status, and having autonomy of their health care which significantly increased the rate of overweight and obesity compared with other women. With these variables combined, the risk of overweight and obesity was increased by 3.3 and 5.4 folds, respectively. The present study identified community-level factors of the region of residence. The contribution of these factors to higher BMI status showed significant regional variation, which was consistent with previous reports of a variation in the nutritional status between clusters [2,35,36]. Reproductive age women living in Dhaka, Mymensingh, Rajshahi, Rangpur, and Sylhet, particularly those in the rural communities, were different from the urban women in their nutritional status. This finding suggests that such community differences should be considered when designing nutritional intervention programs for Bangladeshi women. Policy reforms such as physical, social, and economic that target nutritional improvements should be community-based [36].

Another finding of this study was the higher prevalence of overweight and obesity among women from families with a higher household wealth index and a higher prevalence of underweight among those with a lower household wealth index. These associations were consistent in the unadjusted and adjusted analysis. Such a strong association supports the findings of previous studies in Bangladesh [1,37,38] and elsewhere [39,40,41], which showed that consumption of higher fat and energy-dense food products was tied to higher income. People from higher-income families simply have more money to buy food, have a sedentary lifestyle and need less work, and live in a more affluent region where there is more food security. Dietary factors may have contributed to an increase in the prevalence of overweight and obesity among wealthier women in Bangladesh [42], as was also reported in Nigeria [43]. Studies carried out in developed countries, such as the USA, found a higher prevalence of overweight/obesity among populations with a lower socioeconomic status [44]. However, it remains to be investigated whether or not this situation exists within South Asian countries.

The woman’s age was another risk factor for higher BMI in this study. Older women (aged 35–49 years) had a higher risk of obesity compared with younger women (aged 15–24 years). Similar reports were shown in other studies conducted in Bangladesh and Ethiopia [37,40]. The findings may indicate the lack of physical activity and the high consumption of more energy-dense foods at old age, resulting in obesity [13]. Furthermore, body fat increases and fat-free mass decreases are possible in women over 30 years old. This may offer a possible explanation and may show the link between a higher age and body composition changes [14].

Educated women in Bangladeshi who participated in this study are more likely to have a higher body mass index than non-educated women (i.e., those with no formal education). Our findings are in agreement with a previous study from a developing country, which found a significant association between higher education and obesity compared with women with lower education [15]. As was suggested in a previous study [17], educated women in Bangladesh tend to work in sedentary occupations that are characterized by long hours of sitting or standing at work, which increases their risk of obesity. However, an Iranian study found a lower risk of obesity among women who had more than 12 years of education compared with lesser years of education [45]. The reason for this may be related to the declining physical activity level in people with higher socioeconomic status. However, evidence on the relationship between physical activity and socioeconomic status among reproductive-age women is limited. Even when people were engaged in light activities such as standing or walking around at home (which probably reflects household work) or brisk walking, their risk of obesity and diabetes was significantly reduced, as shown in a previous study [17]. For this reason, it has been recommended that about 30% of obesity cases and 43% of type 2 diabetes cases can be potentially prevented by following a relatively active lifestyle (<10 h/week TV watching and ≥30 min/day of brisk walking) [17].

Women who neither watched the TV nor read the newspaper/magazine had a higher risk of being underweight, which is consistent with studies from elsewhere where sedentary behaviors such as TV watching were associated with a higher risk of overweight and obesity [17]. In that study, for each 2-hr per day increment in TV watching, the authors found an increase in the risk of obesity of about 23% (95% CI: 17–30%) and that for diabetes, the risk was increased by 14% (95% CI: 5–23%) [17]. TV watching reduces physical activity and energy expenditure [17,19]. This is because, as people watch much TV, their food and total energy intake increase. They tend to eat while watching TV and are more likely to follow an unhealthy eating pattern [16]. Those engaged in this behavior are increasingly exposed to unhealthy items marketed on the TV and are more likely to consume these items [18,46]. These may lead to an increase in obesity and diabetes risks [47].

In this study, we found a significant association between contraceptives use and the risk of being underweight in this population. Those who never used contraceptives were more likely to report being underweight which is consistent with previous reports [40,48]. The prevalence of underweight, overweight, and obesity and their related socio-demographic and lifestyle factors among adult women in Myanmar, 2015–2016 [40] showed that women who used contraceptives were less likely to be underweight [48] but more likely to report a higher risk of obesity, especially if used on a regular basis and over a period exceeding two years [8]. 

### Strengths and Limitations

Our study had several strengths. First, the BDHS is a nationally representative survey, which used standardized methods to achieve a high response rate (99%) and has a huge sample size. Second, the use of appropriate adjustments for the complex sampling design of the BDHS provides robust evidence of the factors associated with overweight and obesity in this study population. Third, by using the most recent BDHS data, we have provided additional follow-up information on the present state of the obesity epidemic, which can be used to assess the impact of new policies designed to improve the nutritional state of women. The study had some limitations as well, which should be considered when interpreting the results. First, variables available to measure the individual-, household- and community-level factors were limited. Second, the cross-sectional study design makes it impossible to establish the causal factors for obesity and overweight, simply indicating association. Third, due to the cross-sectional nature of the survey, there was no information on physical activity and women’s dietary information, which are crucial to adjust during regression modeling. 

## 5. Conclusions

The prevalence of overweight and obesity among reproductive-age women in Bangladesh continues to rise. This calls for urgent attention, including a regional public health policy to combat the rising epidemic. Educational campaigns through the media to curb the growing burden of overweight and obesity in this population should target older, educated, and more affluent women and those living in urban areas. The expected cost of overweight or obesity is a burden not only for individuals and families but also for the country. More in-depth research is required to address the challenge of overweight and obesity, considering various critical elements such as nutritional status, physical activity level, and maternal knowledge-associated indicators.

## Figures and Tables

**Figure 1 nutrients-13-04408-f001:**
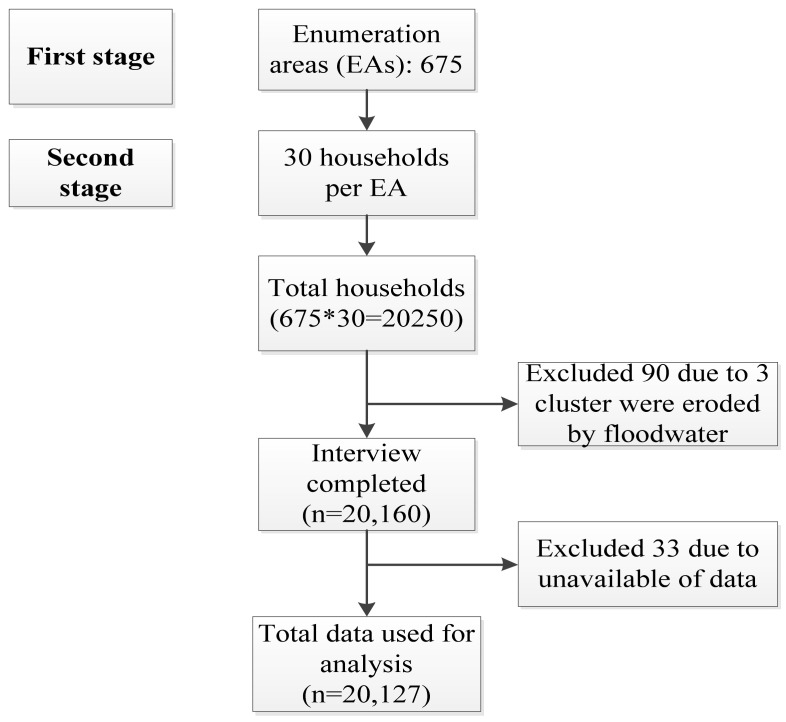
Sampling frame, BDHS2017–2018.

**Figure 2 nutrients-13-04408-f002:**
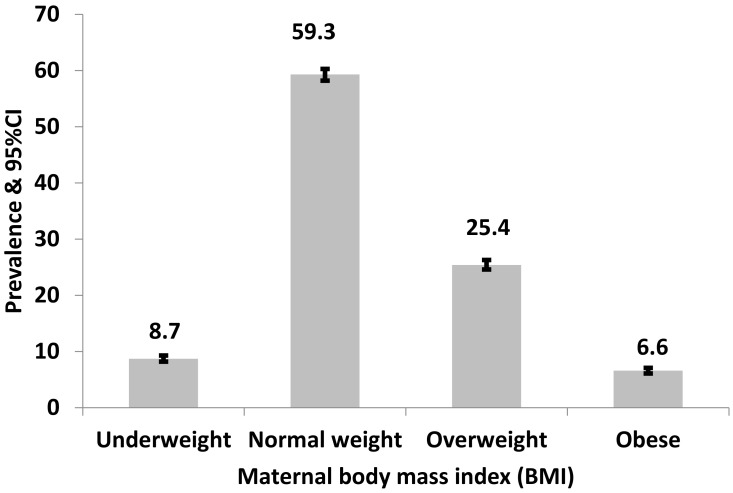
Prevalence and 95% confidence intervals of body mass index categories among women aged 15–49 years in Bangladesh (2004 to 2017).

**Table 1 nutrients-13-04408-t001:** Distribution and prevalence of underweight, normal, overweight, and obesity by individual-, household- and community-level characteristics in Bangladesh between 2017–2018 (*n* = 20,127, except where indicated).

Characteristics	*n* (%)	Underweight	Normal	Overweight	Obesity	
		% (95% CI)	% (95% CI)	% (95% CI)	% (95% CI)	*p**
**Parental level factors**						
**Working status**						
Non-working	10,554 (52.4)	8.1 (4.5–8.7)	56.3 (54.9–57.7)	27.7 (26.5–29.0)	7.9 (7.1–8.6)	<0.001
Working (past 12 months)	9573 (47.6)	9.4 (8.7–10.2)	62.5 (61.3–63.6)	22.9 (21.9–24.0)	5.2 (4.7–5.7)	
**Education**						
No education	3333 (16.6)	11.4 (10.3–12.7)	63.6 (61.7–65.4)	21.0 (19.3–22.8)	4.0 (3.4–4.9)	
Primary	6290 (31.2)	9.4 (8.6–10.2)	60.1 (58.6–61.6)	24.5 (23.2–25.8)	6.0 (5.3–6.8)	<0.001
Secondary and above	10,504 (52.2)	7.5 (6.8–8.2)	57.4 (56.0–58.7)	27.4 (26.3–28.6)	7.7 (7.0–8.5)	
**Father’s education (*n* = 18,983)**						
No education	4130 (21.8)	10.7 (9.6–11.8)	63.7 (62.0–65.4)	21.2 (19.8–22.7)	4.3 (3.7–5.1)	
Primary	6080 (32.0)	9.8 (8.9–10.7)	61.7 (60.1–63.3)	22.9 (21.5–24.3)	5.6 (4.9–6.3)	<0.001
Secondary and above	8773 (46.2)	6.7 (6.1–7.3)	54.9 (53.5–56.3)	29.9 (28.8–31.1)	8.5 (7.7–9.3)	
**Literacy (*n* = 20,117)**						
Cannot read at all	5364 (26.7)	11.0 (10.1–12.0)	62.4 (60.9–63.9)	21.8 (20.5–23.2)	4.8 (4.1–5.5)	<0.001
Able to read only part of sentence	14,753 (73.3)	7.9 (7.3–8.4)	58.1 (56.9–59.3)	26.8 (25.8–27.8)	7.2 (6.7–7.9)	
**Age**						
15–24 years	4882 (24.2)	13.3 (12.3–14.5)	69.3 (67.8–70.8)	14.7 (13.6–16.0)	2.6 (2.1–3.2)	
25–34 years	7785 (38.7)	6.9 (6.2–7.6)	58.2 (56.7–59.7)	27.9 (26.7–29.3)	6.9 (6.2–7.7)	<0.001
35–49 years	7460 (37.0)	7.6 (6.9–8.4)	53.7 (52.2–55.2)	29.8 (28.5–31.2)	8.8 (8.0–9.7)	
**Women status**						
Adolescence (15–19 years)	2063 (10.3)	16.7 (14.9–18.6)	7.8 (7.3–8.3)	10.2 (8.8–11.8)	1.0 (0.7–1.7)	<0.001
Non adolescence (20–49 years)	18,064 (89.7)	72.0 (69.8–74.2)	57.8 (56.7–58.9)	27.2 (26.3–28.1)	7.2 (6.7–7.8)	
**Marital status**						
Currently married	18,984 (93.3)	8.5 (8.0–9.1)	59.1 (58.0–60.1)	25.8 (24.9–26.7)	6. 6 (6.1–7.2)	<0.001
Formerly married ^†^	1143 (5.7)	11.7 (9.9–13.8)	62.5 (59.3–65.6)	20.1 (17.6–22.9)	5.6 (4.3–7.3)	
**Births in last 5 years**						
No birth	12,517 (62.2)	7. 9 (7.3–8.5)	56.9 (55.8–58.1)	27.9 (26.8–28.9)	7.3 (6.7–7.9)	<0.001
Yes	7610 (37.8)	10.1 (9.3–10.9)	63.1 (61.6–64.5)	21.4 (20.2–22.6)	5.5 (4.8–6.1)	
**Currently breastfeeding**						
Yes	4138 (20.6)	12.2 (11.2–13.3)	7.8 (7.3–8.4)	15.9(14.6–17.3)	3.9 (3.2–4.7)	<0.001
No	15,989 (79.4)	67.9 (66.3–69.7)	67.9 (66.3–69.7)	27.9 (27.0–28.9)	7.28 (6.7–7.9)	
**Contraceptive use**						
Use	11,742 (58.3)	8.1 (7.5–8.8)	59.7 (58.6–60.9)	25.6 (24.7–26.6)	6.5 (5.9–7.1)	0.6129
Not using	8385 (41.7)	9.6 (8.8–10.3)	58.6 (57.2–60.0)	25.1 (23.8–26.4)	6.7 (6.0–7.4)	
**Reads newspaper or magazine**						
At least once a week	648 (3.2)	2.8 (1.8–4.4)	42.0 (37.3–46.9)	38.5 (34.2–43.0)	16.7 (13.7–20.1)	
Less than once a week	1218 (6.1)	4.6 (3.4–6.1)	54.2 (51.0–57.3)	30.3 (27.6–33.2)	10.9 (9.1–13.0)	<0.001
Never	18,261 (90.7)	9.2 (8.7–9.7)	60.2 (59.2–61.2)	24.7 (23.8–25.6)	6.0 (5.5–6.4)	
**Listens to radio (*n* = 20,125)**						
At least once a week	414 (2.0)	8.3 (5.7–11.9)	61.7 (56.2–66.8)	24.9 (20.5–29.9)	5.1 (3.2–8.0)	
Less than once a week	561 (2.6)	8.6 (5.8–11.1)	57.4 (52.0-61.8)	26.9 (22.9–31.3)	7.7 (5.7–10.3)	<0.001
Never	19,150 (95.2)	8.7 (8.2–9.3)	59.3 (58.2–60.3)	25.4 (24.5–26.3)	6.6 (6.1–7.1)	
**Watches television**						
At least once a week	11,061 (54.9)	6.3(5.7–6.9)	54.7 (53.3–56.0)	29.8 (28.7–30.9)	9.3 (8.5–10.1)	
Less than once a week	1842 (9.2)	10.6(9.1–12.3)	62.7 (60.3–65.1)	22.2 (20.1–24.5)	4.4 (3.4–5.8)	<0.001
Never	7224 (35.9)	11.9 (11.0–12.9)	65.3 (64.0–66.6)	19.7 (18.6–20.8)	3.1 (2.6–3.5)	
**Power over earnings**						
Husband alone	7119 (35.4)	9.9 (9.1–10.8)	61.5 (60.1–62.39)	22.7 (21.5–23.9)	5.8 (5.2–6.5)	<0.001
Woman alone or joint decision	13,008 (64.6)	8.1 (7.5–8.7)	58.0 (56.8–59.2)	26.9 (25.9–28.0)	7.0 (6.4–7.6)	
**Autonomy Health Care**						
Husband alone	5617 (27.9)	10.8 (9.8–11.8)	62.5 (61.0–64.0)	21.7 (20.5–23.7)	4.9 (4.3–5.6)	<0.001
Woman alone or joint decision	14,510 (72.1)	7.9 (7.4–8.5)	58.0 (56.8–59.2)	26.9 (25.9–27.9)	7.2 (6.6–7.8)	
**Power over Household Decision making**						
Husband alone	3405 (16.9)	11.6 (10.4–12.9)	63.5 (61.7–65.3)	20.1 (18.6–21.7)	4.8 (4.0–5.6)	<0.001
Woman alone or joint decision	16,722 (83.1)	8.1 (7.6–8.7)	58.4 (57.3–59.5)	26.5 (25.6–27.5)	6.9 (6.4–7.5)	
**Wife beaten for refusing sex**						
Yes	634 (3.2)	10.6 (8.3–13.6)	60.7 (55.9–65.2)	22.84 (19.4-26.7)	5.7 (4.1–8.3)	0.252
No	19,493 (96.8)	8.7 (8.2–9.2)	59.2 58.2–60.2)	25.5 (24.6-26.4)	6.6 (6.1–7.1)	
**Attitudes to domestic Violence**						
Yes	4048 (20.1)	10.1 (9.0–11.3)	60.1 (58.1–62.2)	23.9 (22.2–25.6)	5.9 (5.0–7.0)	0.0043
No	16,079 (79.9)	8.4 (7.8–8.9)	59.1 (57.9–60.2)	25.8 (24.9–26.8)	6.7 (6.2–7.3)	
**Household level factors**						
**Household Wealth Index**						
Poorest	3743 (18.6)	14.9 (13.5–16.3)	68.9 (67.1–70.6)	14.5 (13.3–15.7)	1.8 (1.4–2.3)	
Poorer	3957 (19.7)	11.6 (10.5–12.9)	66.4 (64.6–68.1)	19.0 (17.5–20.6)	3.0 (2.4–3.7)	<0.001
Middle	4059 (20.2)	7.9 (7.0–8.9)	61.9 (60.2–63.7)	25.1 (23.5–26.8)	5.0 (4.3–5.7)	
Richer	4184 (20.7)	6.6 (5.7–7.5)	56.9 (55.1–58.7)	29.3 (27.7–31.0)	7.2 (6.4–8.3	
Richest	4184 (20.8)	3.2 (2.6–3.9)	43.2 (41.2–45.2)	38.2 (36.4–40.0)	15.4 (14.1–16.9)	
**Source of drinking water**						
Unimproved	2197 (10.9)	9.1 (7.7–10.6)	61.2 (58.5–63.9)	23.6 (21.3–26.0)	6.1 (5.0–7.5)	<0.001
Improved	17,930 (89.1)	8.7 (8.2–9.2)	59.0 (57.9–60.1)	25.7 (24.7–26.6)	6.6 (6.1–7.2)	
**Community level factors**						
**Residence**						
Urban	5729 (28.5)	6.2 (5.5–7.0)	50.8 (49.0–50.6)	32.2 (30.7–33.8)	10.8 (9.7–11.9)	<0.001
Rural	14,398 (71.5)	9.7 (9.1–10.4)	62.6 (61.4–63.7)	22.8 (21.8–23.8)	4.9 (4.5–5.5)	
**Geographical region (*n* = 20,126)**						
Barishal	1125 (5.6)	7.6 (6.2–9.2)	61.1 (58.6–63.5)	26.6 (24.4–28.8)	4.7 (3.7–6.0)	
Chattogram	3622 (18.0)	5.9 (4.9–7.1)	56.3 (53.8–58.8)	29.7 (27.6–31.9)	8.1 (6.9–9.5)	
Dhaka	5123 (25.5)	7.0 (6.0–8.2)	55.9 (53.2–58.6)	28.3(26.1–30.7)	8.7 (7.4–10.2)	<0.001
Khulna	2336 (11.6)	8.2 (6.9–9.5)	57.2 (54.5–59.9)	27.3 (25.1–29.6)	7.3 (6.2–8.7)	
Mymensingh	1546 (7.7)	13.1 (11.3–15.1)	63.9 (61.3–66.4)	19.4 (15.2–21.7)	3.7 (2.6–5.3)	
Rajshahi	2802 (13.9)	9.43 (8.2–10.8)	61.1 (58.6–63.4)	23.8 (21.7–26.1)	5.7 (4.6–6.9)	
Rangpur	2380 (11.8)	10.1 (8.5–11.9)	65.5 (62.9–67.9)	19.9 (17.9–22.2)	4.5 (3.5–5.8)	
Sylhet	1192 (5.9)	16.3 (14.4–18.6)	62.0 (58.8–65.2)	18.3 (15.8–21.1)	3.3 (2.4–4.6)	

* = result of the comparison between the categories of body mass index. ^†^ = women who were divorced, separated or widowed. CI = confidence interval.

**Table 2 nutrients-13-04408-t002:** Survey logistic modeling of reproductive-age women being underweight: unadjusted (OR) and adjusted Odds Ratios (aOR), Bangladesh 2017–2018.

Characteristic	Underweight (*n* = 2367)					
	Unadjusted			Adjusted		
	OR	95% CI	*p*-Value	AOR	95% CI	*p*-Value
**Maternal education**						
No education	1.00			1.00		
Primary	0.80	0.69, 0.93	0.004	0.80	0.68, 0.94	0.007
Secondary and above	0.63	0.54, 0.72	<0.001	0.76	0.65, 0.90	0.001
**Watches television**						
At least once a week	1.00			1.00		
Less than once a week	1.77	1.45, 2.16	<0.001	1.29	1.05, 1.59	0.015
Never	2.02	1.79, 2.30	<0.001	1.23	1.06, 1.43	0.006
**Currently breastfeeding**						
No	1.00			1.00		
Yes	0.61	0.54, 0.68	<0.001	0.68	0.60, 0.76	<0.001
**Women status**						
Adolescence (15–19 years)	1.00			1.00		
Non adolescence (20–49 years)	0.42	0.37, 0.49	<0.001	0.44	0.37, 0.51	<0.001
**Marital status**						
Currently married	1.00			1.00		
Formerly married ^†^	1.42	1.16, 1.73	0.001	1.31	1.05, 1.64	0.017
**Contraceptive use**						
Using	1.00			1.00		
Not using	1.19	1.07, 1.33	0.001	1.17	1.04, 1.32	0.008
**Division**						
Barishal	1.00			1.00		
Chattogram	0.77	0.58, 1.02	0.068	1.00	0.76, 1.31	0.973
Dhaka	0.92	0.70, 1.22	0.575	1.42	1.08, 1.88	0.013
Khulna	1.08	0.83, 1.42	0.558	1.41	1.08, 1.84	0.013
Mymensingh	1.83	1.40, 2.41	<0.001	1.86	1.44, 2.41	<0.001
Rajshahi	1.27	0.98, 1.65	0.074	1.50	1.17, 1.93	0.002
Rangpur	1.37	1.03, 1.81	0.032	1.33	1.01, 1.75	0.045
Sylhet	2.38	1.83, 3.09	<0.001	2.67	2.07, 3.45	<0.001
**Wealth Index**						
Richest	1.00			1.00		
Richer	2.11	1.63, 2.73	<0.001	1.89	1.46, 2.46	<0.001
Middle	2.60	2.00, 3.37	<0.001	2.28	1.74, 2.97	<0.001
Poorer	3.94	3.11, 4.99	<0.001	3.11	2.40, 4.04	<0.001
Poorest	5.25	4.12, 6.69	<0.001	3.86	2.94, 5.07	<0.001

^†^ = women who were divorced, separated, or widowed. CI = confidence interval.

**Table 3 nutrients-13-04408-t003:** Survey logistic modeling of reproductive-age women being overweight: unadjusted (OR) and adjusted Odds Ratios (aOR), Bangladesh 2017–2018.

Characteristic	Overweight (*n* = 5164)					
	Unadjusted			Adjusted		
	OR	95% CI	*p*-Value	AOR	95% CI	*p*-Value
**Maternal education**						
No education	1.00			1.00		
Primary	1.22	1.09, 1.38	0.001	1.19	1.06, 1.34	0.004
Secondary and above	1.43	1.27, 1.60	<0.001	1.18	1.05, 1.33	0.006
**Watches television**						
At least once a week	1.00			1.00		
Less than once a week	0.67	0.59, 0.77	<0.001	0.89	0.77, 1.02	0.106
Never	0.58	0.53, 0.63	<0.001	0.90	0.81, 0.99	0.031
**Currently breastfeeding**						
No	1.00			1.00		
Yes	2.05	1.85, 2.26	<0.001	1.91	1.72, 2.12	<0.001
**Women status**						
Adolescence (15–19 years)	1.00			1.00		
Non adolescence (20–49 years)	3.29	2.79, 3.87	<0.001	3.14	2.64, 3.72	<0.001
**Marital status**						
Currently married	1.00			1.00		
Formerly married ^†^	0.73	0.62, 0.86	<0.001	0.76	0.63, 0.92	<0.001
**Power over earnings**						
Husband alone	1.00			1.00		
Woman alone or joint decision	1.25	1.16, 1.35	<0.001	1.16	1.06, 1.26	0.001
**Division**						
Barishal	1.00			1.00		
Chattogram	1.17	1.00, 1.36	0.048	0.89	0.78, 1.03	0.111
Dhaka	1.09	0.93, 1.28	0.274	0.74	0.64, 0.86	<0.001
Khulna	1.04	0.88, 1.22	0.658	0.85	0.74, 0.99	0.033
Mymensingh	0.66	0.55, 0.80	0.000	0.65	0.55, 0.76	<0.001
Rajshahi	0.87	0.73, 1.02	0.091	0.77	0.66, 0.89	<0.001
Rangpur	0.69	0.58, 0.82	<0.001	0.73	0.62, 0.86	<0.001
Sylhet	0.62	0.50, 0.76	<0.001	0.53	0.44, 0.63	<0.001
**Wealth Index**						
Richest	1.00			1.00		
Richer	0.67	0.60, 075	<0.001	0.73	0.65, 0.81	<0.001
Middle	0.54	0.49, 0.61	<0.001	0.59	0.52, 0.67	<0.001
Poorer	0.38	0.33, 0.43	<0.001	0.45	0.39, 0.52	<0.001
Poorest	0.27	0.24, 0.31	<0.001	0.34	0.29, 0.40	<0.001

^†^ = women who were divorced, separated, or widowed. CI = confidence interval.

**Table 4 nutrients-13-04408-t004:** Survey logistic modeling of reproductive-age women being obese: unadjusted (OR) and adjusted Odds Ratios (aOR), Bangladesh 2017–2018.

Characteristic	Obesity (*n* = 1362)					
	Unadjusted			Adjusted		
	OR	95% CI	*p*-Value	AOR	95% CI	*p*-Value
**Maternal education**						
No education	1.00			1.00		
Primary	1.53	1.22, 1.91	<0.001	1.58	1.25, 1.99	<0.001
Secondary and above	1.99	1.61, 2.46	<0.001	1.50	1.16, 1.94	<0.001
**Watches television**						
At least once a week	1.00			1.00		
Less than once a week	0.45	0.34, 0.61	<0.001	0.72	0.54, 0.97	0.029
Never	0.31	0.26, 0.37	<0.001	0.63	0.51, 0.77	<0.001
**Currently breastfeeding**						
No	1.00			1.00		
Yes	1.94	1.61, 2.35	<0.001	1.33	1.09, 1.63	<0.001
**Women status**						
Adolescence(15–19 years)	1.00			1.00		
Non adolescence (20–49 years)	7.02	4.40, 11.19	<0.001	3.26	1.93, 5.49	<0.001
**Age**						
15–24 years	1.00			1.00		
25–34 years	2.76	2.19, 3.48	<0.001	1.94	1.49, 2.52	<0.001
35–49 years	3.57	2.85, 4.48	<0.001	2.73	2.10, 3.56	<0.001
**Autonomy Health Care**						
Husband alone	1.00			1.00		
Woman alone or joint decision	1.50	1.29, 1.73	<0.001	1.25	1.08, 1.45	0.003
**Working status**						
Non-working	1.00			1.00		
Working (past 12 months)	0.64	0.56, 0.73	<0.001	0.74	0.64, 0.85	<0.001
**Reads newspaper or magazine**						
At least once a week	1.00			1.00		
Less than once a week	0.61	0.46, 0.82	0.001	0.89	0.66, 1.19	0.439
Never	0.32	0.25, 0.40	<0.001	0.75	0.59, 0.95	0.017
**Division**						
Barishal	1.00			1.00		
Chattogram	1.77	1.31, 2.39	<0.001	1.14	0.87, 1.49	0.332
Dhaka	1.91	1.41, 2.59	<0.001	1.03	0.79, 1.35	0.813
Khulna	1.59	1.16, 2.17	0.004	1.15	0.88, 1.51	0.308
Mymensingh	0.77	0.49, 1.20	0.248	0.74	0.50, 1.08	0.121
Rajshahi	1.20	0.87, 1.67	0.266	1.00	1.75, 1.33	0.992
Rangpur	0.96	0.67, 1.37	0.805	1.07	0.78, 1.47	0.679
Sylhet	0.69	0.45, 1.05	0.082	0.51	0.37, 0.71	<0.001
**Wealth Index**						
Richest	1.00			1.00		
Richer	0.43	0.36, 0.51	<0.001	0.52	0.44, 0.62	<0.001
Middle	0.29	0.24, 0.34	<0.001	0.37	0.30, 0.46	<0.001
Poorer	0.17	0.13, 0.21	<0.001	0.26	0.20, 0.35	<0.001
Poorest	0.10	0.07, 0.13	<0.001	0.19	0.13, 0.27	<0.001

CI = confidence interval.

## Data Availability

The study was based on an analysis of existing survey dataset that are available to apply for online, with all identifier information removed. Written informed consent for the present analysis was not necessary because secondary data analysis did not involve interaction with the participants. The data collection methods for the 2017–2018 BDHS data used in this analysis, including the consent process, have been previously described [29]. Written informed consent for the present analysis was not necessary because secondary data analysis did not involve interaction with the participants. This study was based on a public domain dataset that is freely available online: https://dhsprogram.com/data/dataset/Bangladesh_Standard-DHS_2017.cfm?flag=0 (accessed on 2 September 2021).

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
