# Peer review of "Underweight, Overweight and Obesity among Reproductive Bangladeshi Women: A Nationwide Survey"

_nutrients, 2021, doi:10.3390/nu13124408_

Round 1

Reviewer 1 Report

This is a survey study including 20127 female participants. Various health characteristics were recorded and compared among BDHS-2014. Stratifications were made regarding different sociodemographic characteristics and health status. They found that the overweight and obesity was higher among urban residence, educated and affluent women.

This is an interesting study with some new findings in this area of research. The sample size of subjects is enough for analysis. However, I nevertheless have the following comments that required to be addressed.

  1. The statistical methods adopted ordinal logistic regression and described should be made specific to the research question. In fact, I am not familiar to ordinal logistic regression. What difference between ordinal logistic regression and conventional logistic regression?

  1. For Figure 1, I suggested not to be redundant. The authors should clarify this concern.

  1. How does to the post-hoc power performed by this study? And comparison between sociodemographic groups for more meaningful interpretation. Which variables or groups used post-hoc comparison?

  1. Please brief the dietary pattern of participants. For example, The Western pattern diet. The authors should clarify this concern to add readability.

  1. Any study involved health belief theory for BDHS-2017-18?

  1. Does any bias exist in cross-sectional study? The authors should clarify this concern.

Author Response

Responses to Reviewer 1

Comments and Suggestions for Authors

This is a survey study including 20127 female participants. Various health characteristics were recorded and compared among BDHS-2014. Stratifications were made regarding different sociodemographic characteristics and health status. They found that the overweight and obesity was higher among urban residence, educated and affluent women.

This is an interesting study with some new findings in this area of research. The sample size of subjects is enough for analysis. However, I nevertheless have the following comments that required to be addressed.

  1. The statistical methods adopted ordinal logistic regression and described should be made specific to the research question. In fact, I am not familiar to ordinal logistic regression. What difference between ordinal logistic regression and conventional logistic regression?

Responses: We have revised the analysis and all analyses are now in logistic regression as suggested by the reviewer.

  1. For Figure 1, I suggested not to be redundant. The authors should clarify this concern.

  Responses: Thank you very much for the comments. Figure 1 now revised.

  1. How does to the post-hoc power performed by this study? And comparison between sociodemographic groups for more meaningful interpretation. Which variables or groups used post-hoc comparison?

 Response: Thank you very much for the comments. We have revised the analysis and above mentioned comments are no longer relevant.

  1. Please brief the dietary pattern of participants. For example, The Western pattern diet. The authors should clarify this concern to add readability.

Response: Women dietary information was not collected except breastfeeding status. However, we have reflected dietary as one of the limitations of the study.

  1. Any study involved health belief theory for BDHS-2017-18?

  Response: The survey does not include qualitative or such information.

  1. Does any bias exist in cross-sectional study? The authors should clarify this concern.

  Response: Thank you very much. These are the limitations of the study, which we have mentioned in the limitation section.

Reviewer 2 Report

Content: This articles reports data derived of 20,127 women aged 15-49 years from the Bangladesh Demographic and Health Survey (BDHS). Demographical data were correlated with the Body Mass index (BMI) to understand the risk factors for underweight, overweight, and obesity. The authors found that the risk of overweight and obese compared to underweight was higher among women with a high level of education, rich households, age of ≥25 years, and women who made decisions over earnings and health care with their husbands. Women exposed to media and those who lived in urban areas were more likely to report overweight and obesity relative to underweight. Having this huge data set from Bangladesh available is of great scientific value.

Major Comments: It is unusual to compare obese and overweight people to underweight women, because this is comparing two extremes with each other. Instead, it would be more informative to to compare normal-weight women with underweight, overweight and obese people.

Thus, all the tables need to show four groups: Women with underweight, normal weight, overweight and obesity. These four groups should be compared and the results should be presented accordingly.

Minor comments: Underweight people might simply not have enough to eat. This may not be a question of education. If discussing potential interventions, education is not the only possible way. Labelling sweet and fatty foods as such or having higher taxes on sweet and fatty foods might also be possible.

The title may be shortened to "Overweight and obesity among reproductive Bangladeshi women: A nationwide survey."

Author Response

Responses to Review2

Comments and Suggestions for Authors

Content: This articles reports data derived of 20,127 women aged 15-49 years from the Bangladesh Demographic and Health Survey (BDHS). Demographical data were correlated with the Body Mass index (BMI) to understand the risk factors for underweight, overweight, and obesity. The authors found that the risk of overweight and obese compared to underweight was higher among women with a high level of education, rich households, age of ≥25 years, and women who made decisions over earnings and health care with their husbands. Women exposed to media and those who lived in urban areas were more likely to report overweight and obesity relative to underweight. Having this huge data set from Bangladesh available is of great scientific value.

Response: Thank you and we appreciate your kind comment.

Major Comments: It is unusual to compare obese and overweight people to underweight women, because this is comparing two extremes with each other. Instead, it would be more informative to to compare normal-weight women with underweight, overweight and obese people.

Response: The manuscript has been amended to reflect the above suggestions.

Thus, all the tables need to show four groups: Women with underweight, normal weight, overweight and obesity. These four groups should be compared and the results should be presented accordingly.

Response: Tables now presented factors associated with underweight, overweight and obesity.

Minor comments: Underweight people might simply not have enough to eat. This may not be a question of education. If discussing potential interventions, education is not the only possible way. Labelling sweet and fatty foods as such or having higher taxes on sweet and fatty foods might also be possible.

Response: Agreed and our analysis showed, and we have suggested possible interventions for Underweight groups.

The title may be shortened to "Overweight and obesity among reproductive Bangladeshi women: A nationwide survey."

Response: Titled revised show reads “Underweight, Overweight and obesity among reproductive Bangladeshi women: A nationwide survey."

Round 2

Reviewer 1 Report

Thanks for your efforts on revision. Please confirm outcome of this study was discrete and nominal.  

Reviewer 2 Report

The authors have worked hard on the manuscript and have considered my suggestions. The article has significantly improved. I have no further comments.